# Correlating continuously captured home-based digital biomarkers of daily function with postmortem neurodegenerative neuropathology

**Nathan C. Hantke**[1,2,3]*, **Jeffrey Kaye**[1,2], **Nora Mattek**[1,2], **Chao-Yi Wu**[1,2,4], **Hiroko H. Dodge**[1,2,4], **Zachary Beattie**[1,2], **Randy Woltjer**[5]

**1** Department of Neurology, Oregon Health & Science University, Portland, OR, United States of America, **2** Oregon Center for Aging & Technology (ORCATECH), Portland, OR, United States of America, **3** Mental Health and Clinical Neuroscience Division, VA Portland Health Care System, Portland, OR, United States of America, **4** Department of Neurology, Massachusetts General Hospital, Harvard Medical School, Boston, MA, United States of America, **5** Department of Pathology and Laboratory Medicine, Oregon Health & Science University, Portland, OR, United States of America

* Hantke@ohsu.edu

**Data Availability Statement:** Data cannot be shared publicly given the sensitive nature of the data. Qualified researchers may obtain access to

## Abstract

### Background

Outcome measures available for use in Alzheimer's disease (AD) clinical trials are limited in ability to detect gradual changes. Measures of everyday function and cognition assessed unobtrusively at home using embedded sensing and computing generated "digital biomarkers" (DBs) have been shown to be ecologically valid and to improve efficiency of clinical trials. However, DBs have not been assessed for their relationship to AD neuropathology.

### Objectives

The goal of the current study is to perform an exploratory examination of possible associations between DBs and AD neuropathology in an initially cognitively intact community-based cohort.

### Methods

Participants included in this study were ≥65 years of age, living independently, of average health for age, and followed until death. Algorithms, run on the continuously-collected passive sensor data, generated daily metrics for each DB: cognitive function, mobility, socialization, and sleep. Fixed postmortem brains were evaluated for neurofibrillary tangles (NFTs) and neuritic plaque (NP) pathology and staged by Braak and CERAD systems in the context of the "ABC" assessment of AD-associated changes.

### Results

The analysis included a total of 41 participants (M±SD age at death = 92.2±5.1 years). The four DBs showed consistent patterns relative to both Braak stage and NP score severity. Greater NP severity was correlated with the DB composite and reduced walking speed.

de-identified data utilized in this study by contacting our centers webpage (https://www.ohsu.edu/alzheimers-disease-research-center/data-resources).

**Funding:** This work was supported by several grants, including the National Institute on Aging: P30AG024978, R01AG024059, P30AG008017, P30AG066518. https://www.nia.nih.gov/.

**Competing interests:** JK in the past 36 months has been directly compensated for serving on a Data Safety Monitoring Committee for Eli Lilly, the Scientific Advisory Board of Sage Bionetworks, the Roche/Genentech Scientific Advisory Committee for Digital Health Solutions in Alzheimer's Disease, and as an external Advisory Committee member for two Alzheimer's Disease Research Centers. He has received research support awarded to his institution (Oregon Health & Science University) from the NIH, NSF, the Department of Veterans Affairs, USC Alzheimer's Therapeutic Research Institute, Merck, AbbVie, Eisai, Green Valley Pharmaceuticals, and Alector. He has received reimbursement through Medicare or commercial insurance plans for providing clinical assessment and care for patients. He has served on the editorial advisory board and as Associate Editor of the journal, Alzheimer's & Dementia and as Associate Editor for the Journal of Translational Engineering in Health and Medicine. HD works as a consultant for Biogen and is supported by the following federal grants: NIH R01AG051628, R01AG056102, R01AG069782, P30AG066518, R01AG072449, P30AG008017, P30AG024978, U2CAG054397, R01AG056712, R01AG0380651, P30AG053760, U01NS100611, U2CAG057441, U01NS106670, R01AG054484, RF1AG072449. ZB and JK have a financial interest in Life Analytics, Inc., a company that may have a commercial interest in the results of this research and technology. This potential conflict of interest has been reviewed and managed by Oregon Health & Science University. Remaining authors have no disclosures. This does not alter our adherence to PLOS ONE policies on sharing data and materials.

Braak stage was associated with reduced computer use time and increased total time in bed.

## Discussion

This study provides the first data showing correlations between DBs and neuropathological markers in an aging cohort. The findings suggest continuous, home-based DBs may hold potential to serve as behavioral proxies that index neurodegenerative processes.

## Introduction

The neurodegenerative disorder Alzheimer's disease (AD) currently affects approximately one in nine persons age 65 years or older in the United States of America, a number that is expected to rise as the current population ages [1]. AD is characterized by a progressive decline in cognitive function, reduction in functional abilities, and neuropathological markers that include neurofibrillary tangles (NFTs) and neuritic plaques (NPs) [2–5].

The expanding science behind AD pathogenesis is promising, but early detection continues to prove complex. Subtle cognitive change and decline in instrumental activities of daily living (IADLs) are often early signals of future dementia [6,7]. Monitoring changes in cognitive status is generally achieved through repeated clinical visits. Episodic clinical assessments such as cognitive screeners often lack sufficient ecological validity to generalize to real-world settings by capturing only one point in time and in a setting that does not indicate how a person functions in his/her daily environment [8–11]. Similarly, IADL questionnaires do not account well for within-person variability, are by their nature subjective, and often do not capture decreased efficiency for completing daily tasks.

Monitoring behavior in the home using remote sensing and digital technologies addresses many of the validity concerns of currently used methods without disrupting usual routines [12,13]. High data capture frequency from passive sensors provide digital biomarkers [DBs], defined as objective, quantifiable physiological and behavioral data collected and measured by means of digital devices [13–15]. There is growing empirical evidence that passive monitoring of daily activities, such as changes in daily computer use, mobility about the home, medication-taking, sleep routines, phone use, and driving, provides insight into every day cognitive function [16–20].

DBs have demonstrated an ability to assess change in daily function over time in older adults who are cognitively intact and in those with clinically diagnosed MCI [13,14], yet there remains a gap in understanding the relationship of these objective functional changes (i.e., DBs) and the underlying brain pathology. A prior cross-sectional study found a significant relationship between less daily computer use and medial temporal lobe atrophy [21], a brain region that is known to be affected early-on in AD pathologically. This finding provided indirect, in vivo evidence of a link between DBs and AD, but did not directly measure the gold standard of post-mortem pathology data [5].

Few autopsy-based studies exist that examine a direct link between objective functional activity measures and underlying neuropathology. Studies have examined the relationship between measured physical activity, cognition, and brain pathology among older adults [22,23]. Another study observed that lower accelerometer measured physical activity was associated with brain pathologies [24]. However, studies related to more complex activities of daily living assessed naturalistically are lacking. With this background in mind, we aimed to determine the association of DBs to AD neuropathology in an initially non-demented,

longitudinally-monitored, community-based cohort. Secondly, we examined the association between objective DBs with antemortem global cognition via Mini Mental State Examination (MMSE), functional status via Functional Activities Questionnaire (FAQ), and AD neuropathology.

## Methods

### Participants

Forty-one participants were included in the analysis. Inclusion criteria at study onset was age 65 years and older, in average health for age without poorly controlled medical illnesses, not demented at study entry (Mini-Mental State Examination [MMSE] scores >24) [25], self-report of being able to use a computer proficiently, and living independently (12). Assessment of baseline health was based upon review of participants' medical history, medication lists, and completion of the modified Cumulative Illness Rating Scale [26,27]. Medical illnesses with the potential to limit physical participation (e.g., wheelchair bound) or likely to lead to death over the course of 35 months (such as certain cancers) were study exclusions. All participants completed annual clinical evaluations, including administration of the Clinical Dementia Rating (CDR) scale [28] at initial and subsequent visits to monitor for the presence of MCI and transition to dementia, and were followed until death. All participants provided written informed consent and had been previously enrolled in ongoing longitudinal studies of aging and in-home monitoring (www.orcatech.org). Participants were recruited from the Portland, Oregon metropolitan area through advertisement and presentations at local retirement communities. The study protocols were approved by the Oregon Health & Science University Institutional Review Board (Life Laboratory (LL) IRB #2765; ISAAC IRB #2353). All procedures involving human participants were in accordance with the ethical standards of the institutional and/or national research committee and with the 1964 Helsinki declaration and its later amendments or comparable ethical standards. All cited articles in this manuscript contain human and/or animal work approved by institutional review boards prior to publication. Additional details of the sensor systems and study protocols have been published elsewhere [12,29]. Data were collected between the years 2008 to 2018. During this time period, 65% of the cohort participants died and went to brain autopsy.

### Digital biomarker activity metrics

All of the participants had an unobtrusive, pervasive technology platform installed in their home consisting of X10 passive infrared (PIR) motion and X10 door contact sensors, and computer use monitoring software (**Fig 1**) (12). Algorithms, run on the continuously-collected passive sensor data from the technology platform, generated daily functional metrics for each participant. From the array of DBs, four measures representing four domains of functioning were selected based on prior research demonstrating differences in these measures during everyday life in those with MCI compared to those with normal cognition, as well as their key roles in gauging functional ability: (1) cognitive function based on frequency of computer use [19]; (2) mobility based on daily mean walking speed [30,31]; (3) socialization based on time out of home [32]; and, (4) duration of time in bed [20].

Computer use was measured by the number of days participants used his/her computer during the past year. Daily mean walking speed (cm/s) was measured using an array of in-home sensors which passively identified how quickly and frequently participants were passing under a sensor line [31,33–36]. Algorithms estimating the speed of walking from the in-home sensor data have been validated against a 'gold standard' gait mat [34,37]. Time out of home (ie., total time spent out of the home per day in hours) was measured using the PIR motion

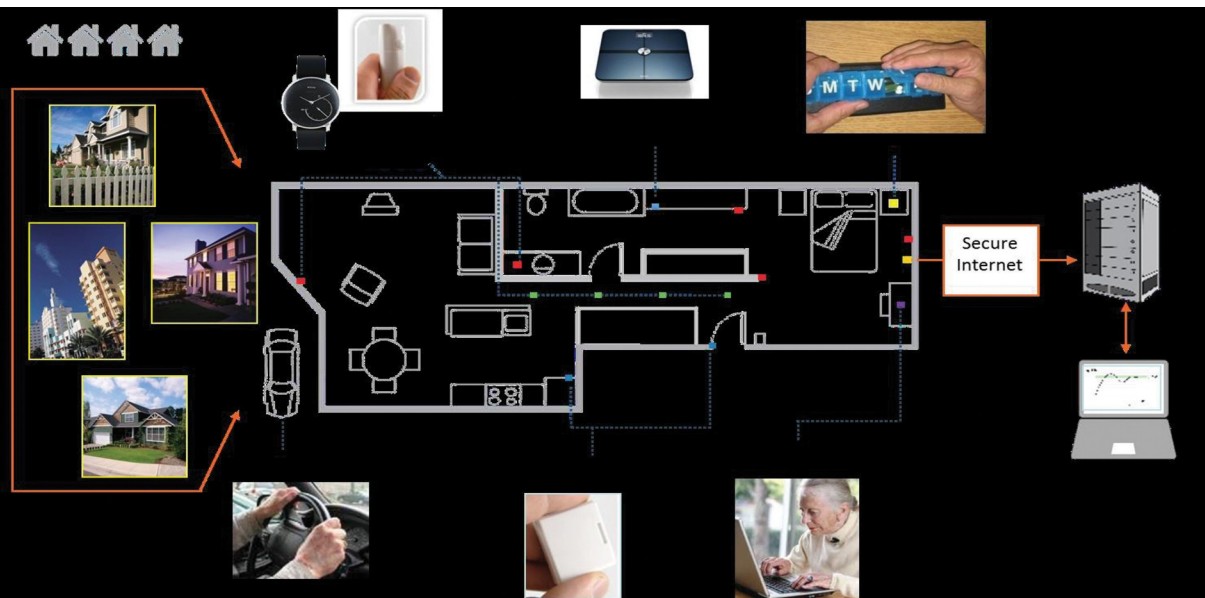

**Fig 1. Home-based pervasive sensor and computing system.**

sensors and door contact sensors, which were able to detect activity (or lack thereof) in the home, and the door openings and closings [32,35]. Data from PIR motion sensors in each room of the home, including the bedroom, were used to measure total time in bed, in hours [38]. Only participants living alone were included in order to clearly disambiguate in-home movement of multiple residents.

## Neuropathologic data

Fixed postmortem brains were evaluated for NFTs and NP pathology and staged by Braak [39] and Consortium to Establish a Registry for AD (CERAD) systems [40]. Brains were fixed in neutral-buffered formaldehyde solution for at least two weeks and examined grossly, as well as microscopically. For microscopic evaluation, tissue samples were taken from all cortical lobes bilaterally or unilaterally, frontal lobe white matter, anterior cingulate gyrus, hippocampus, amygdala, bilateral striatum and thalamus, midbrain, pons, medulla, and cerebellum. Six-micrometer sections were routinely stained with hematoxylin-eosin and Luxol fast blue. Selected sections of hippocampus and neocortical regions were immunostained using PHF1 antibody to tau and additional sections were stained to determine the presence of beta-amyloid (4G8 antibody, Biolegend, San Diego, CA), alpha-synuclein (MJFR1 antibody, Abcam, Waltham, MA), and TDP-43 (1D3 antibody, Biolegend, San Diego, CA). Clinical and pathologic diagnoses were established using current consensus criteria [41–45]. Information related to NP and NFT burdens, amyloid angiopathy, large vessel strokes or lacunes, presence of Lewy bodies (LBs), hippocampal sclerosis (HS), and degree of arteriosclerosis were summarized using the National Alzheimer's Coordinating Center (NACC) Neuropathology Data reporting format [46]. The NACC Neuropathology Form changed versions (versions 9, 10, and 11) over the course of the study, resulting in hippocampal sclerosis data only being available for a limited number of subjects (n = 19).

## Statistical analysis

Summary statistics were generated for participant characteristics and pathologic variables. A normally distributed composite DB measure including the four activity domains (cognition,

mobility, socialization, and sleep/time in bed) was constructed by *z*-score normalizing the four individual domain metrics. Faster walking speed, more time out of home, more days with computer use, and less total time in bed contributed to a higher composite DB score. Data analysis was conducted using the home-monitored data from the 12-month period of available sensor information prior to death to avoid measuring acute, end of life changes in activity.

Differences in DBs according to individual neuropathological categories (e.g., Braak stage, plaque severity) and composite DB score were presented visually as box plots in **Figs 2–4**. Independent t-tests, spearman rank (non-parametric) correlations, and linear regression models were generated when appropriate to examine the association between neuropathological

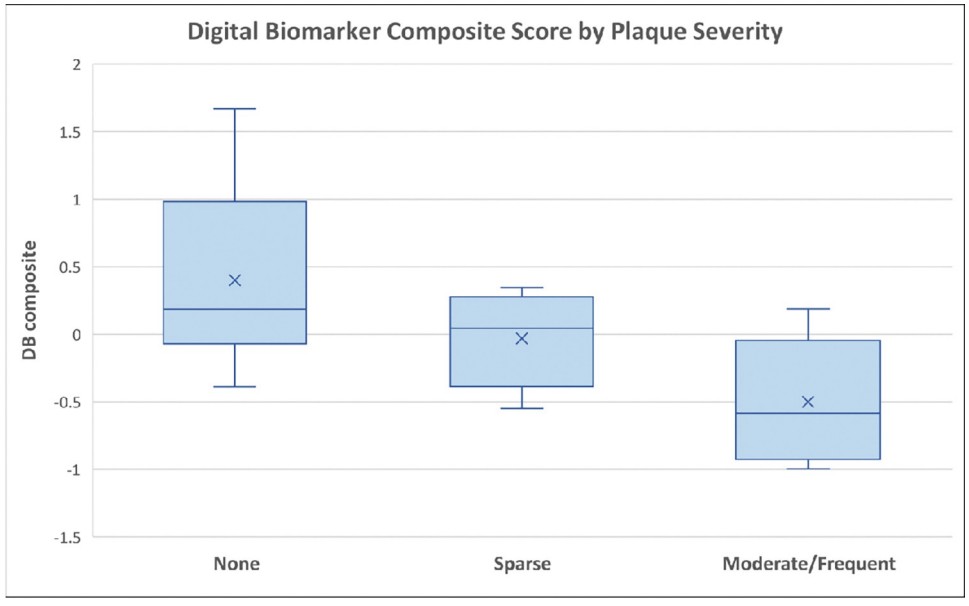

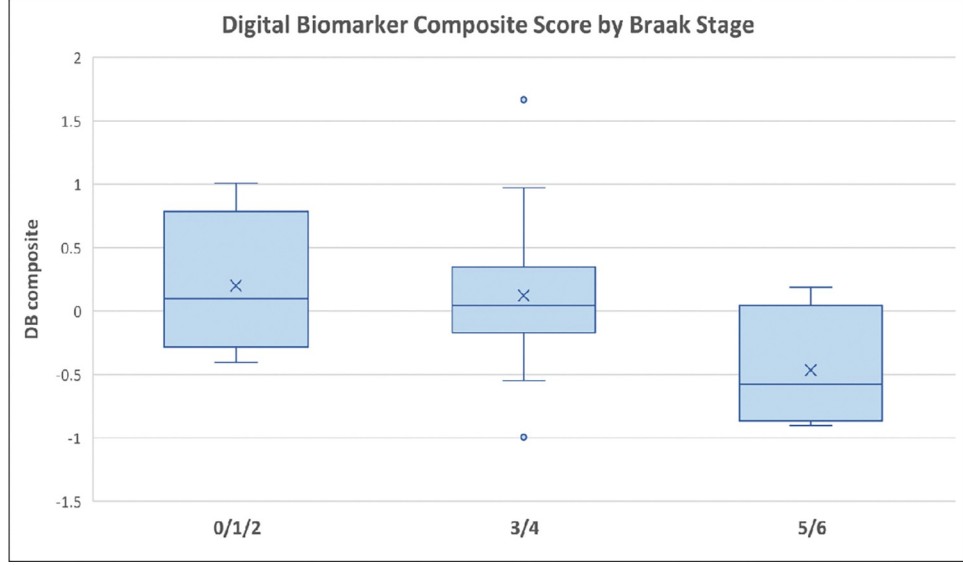

**Fig 2. Distributions of multi-domain activity level by neuritic plaque score and Braak score.** a. *Box plots of the distribution of multi-domain activity level by neuritic plaque score (p = .01).* b. *Box plots of the distribution of multi-domain activity level by Braak score (p = .16).*

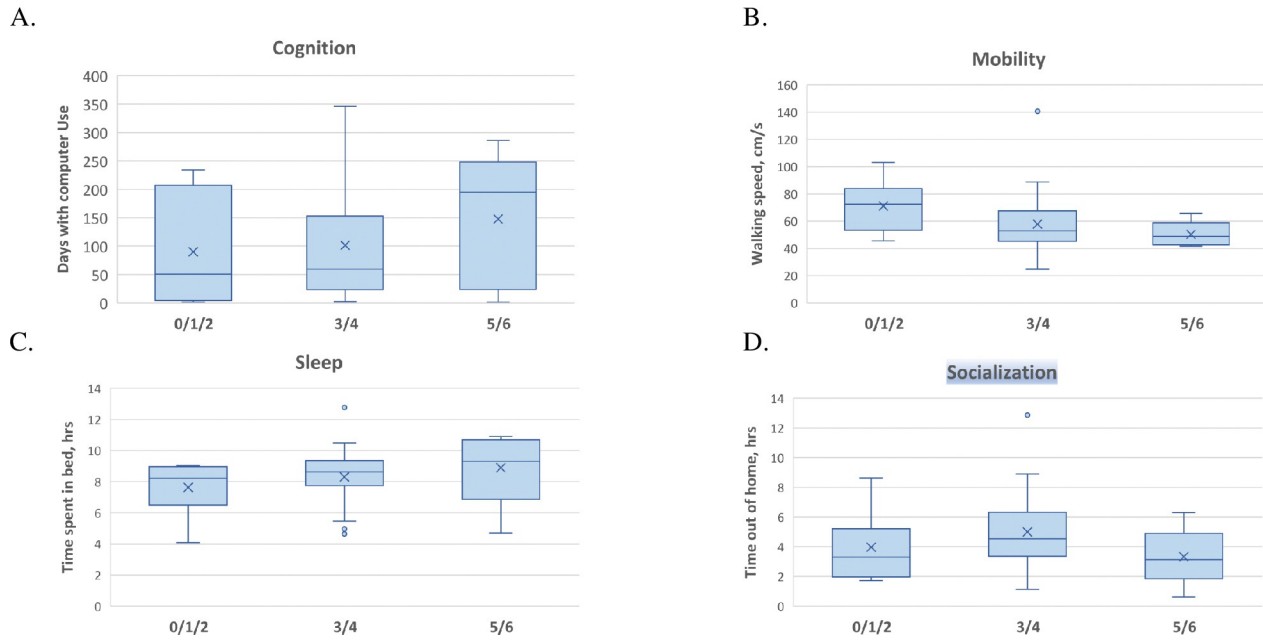

**Fig 3. Digital biomarkers and Braak scores.** (A) Cognition by Braak score, measured in days with Computer use; (B) Mobility by Braak score, measured in *M* walk speed (cm/s); (C) Sleep by Braak score, measured in *M* time spent in bed; (D) Socialization by Braak score, as measured by *M* time out of home (TOH).

categories and the DB composite metric (**Table 2**; **S1 Table**), as well as neuropathological categories, antemortem global cognition (last MMSE before death), and a functional measure (FAQ at last visit before death). Due to a small sample size we were unable to control for covariates. Analyses were performed using SAS software 9.4 (Cary, NC).

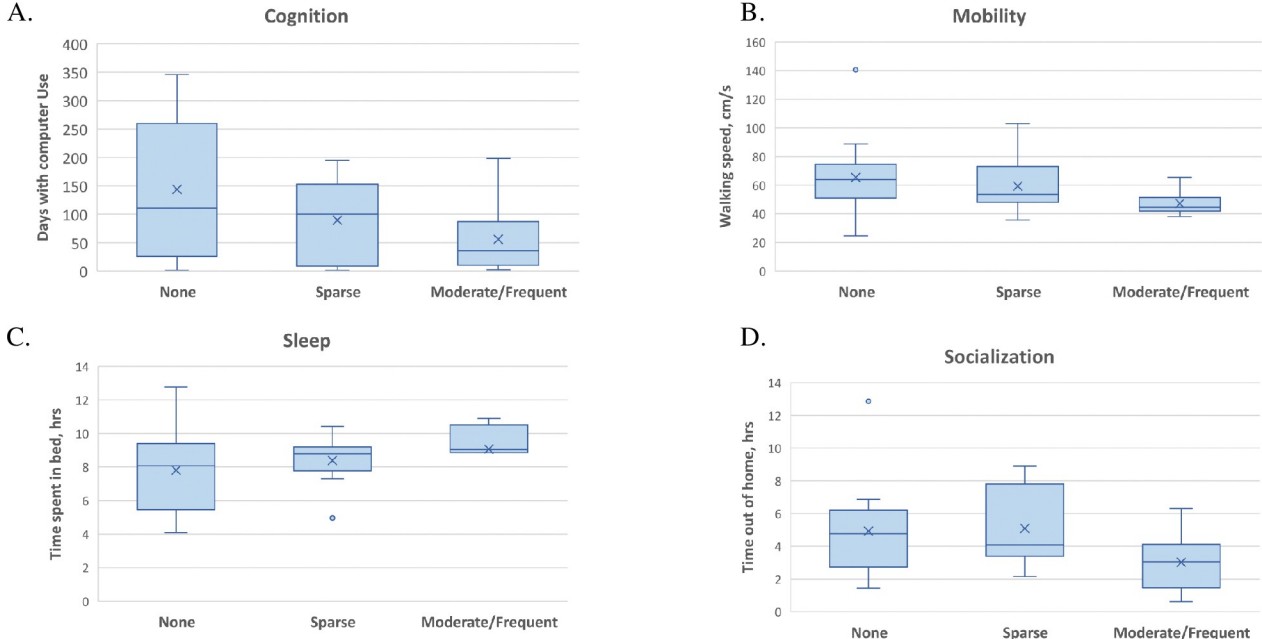

**Fig 4. Digital biomarkers and neuritic plaque scores.** (A) Cognition by plaque score, measured in days with Computer use; (B) Mobility by plaque score, measured in *M* walk speed (cm/s); (C) Sleep by plaque score, measured in *M* time spent in bed (TST); (D) Socialization by plaque score, as measured by *M* time out of home (TOH).

**Table 1. Participant demographic, clinical, and digital biomarker characteristics.**

| Variable | Full sample (*N* = 41) |
|---|---|
| *Demographics* | |
| Age at death, *M (SD)* yrs | 92.2 (5.1) |
| Female, *No. (%)* | 34 (82.9%) |
| Education, *M (SD)* yrs | 15.6 (2.7) |
| *Cognition and functional status* | |
| Clinical diagnosis antemortem | |
| Cognitively unimpaired | 19 (46%) |
| MCI | 16 (39%) |
| Dementia | 6 (15%) |
| MMSE before death, *Median (SD)* | 27.0 (5.9) |
| FAQ before death (n = 26), *Median (SD)* | 1.0 (8.8) |
| Months from last clinical visit to death, *Median (SD)* | 6.5 (13.6) |
| *Neuropathology*, *No. (%)* | |
| Braak stage | |
| I/II | 7 (17.1%) |
| III/IV | 29 (70.7%) |
| V/VI | 5 (12.2%) |
| Neuritic plaque score | |
| None | 16 (39.0%) |
| Sparse | 17 (41.5%) |
| Moderate/frequent | 8 (19.5%) |
| Large vessel stroke/lacunar stroke | 7 (17.1%) |
| Hippocampal sclerosis (n = 19) | 1 (5.3%) |
| Lewy bodies | 3 (7.3%) |
| *Digital biomarker metrics*, *M (SD)* | |
| Time from last sensor recording to death in days *(Median, range)* | 1 (1–2753) |
| Walking speed, cm/s (n = 36) | 59 (22) |
| Days on computer in one year (n = 29) | 107 (103) |
| Time out of home per day, hours (n = 40) | 4.6 (2.5) |
| Time in bed per night, hours (n = 35) | 8.3 (2.0) |
| Composite DB z-score (n = 23) | 0.0 (0.6) |

*Note*. Values are displayed as *Mean* (*SD*) or *Median (SD)* for continuous variables based on normality of distributions and *No.* (%) for all categorical variables.

Abbreviations: MCI, mild cognitive impairment; M, mean; MMSE, Mini-Mental Status Examination; FAQ, Functional Activities Questionnaire.

## Results

Characteristics of the 41 participants are described in **Table 1**. Cohort mean age at death was 92.2 years. Thirty-two percent (n = 13) of participants were ApoE ε4 carriers; the sample size and number of variables included in the analysis did not allow for additional sub analysis including ε4 status. Study participants had sensors in their home for an average of 5.8 years (2.4); median time from last DB home monitoring data and death was one day. Twenty-three participants (56%) died while their home was sensored; median last MMSE score before death was 27.0 (5.9). A subset of participants had each of the four individual DBs available; walking speed (n = 36), time out of home (n = 40), total time in bed (n = 35) and computer use (n = 29). Twenty-three participants (56%) had all four individual DBs available to calculate the

**Table 2. Correlations between digital biomarker activity metrics and postmortem pathology.**

| | Statistics | Walking speed (cm/s) | Number of days with computer use | Time out of homee (hrs) | Total time spent in bed (hrs) | Composite activity measure |
|---|---|---|---|---|---|---|
| **Braak stage** | Coefficient | 0.171 | -0.437 | 0.050 | 0.395 | -0.302 |
| (1–6) | p-value | 0.320 | 0.018 | 0.759 | 0.019 | 0.162 |
| | n | 36 | 29 | 40 | 35 | 23 |
| **Neuritic plaque severity** | Coefficient | -0.379 | -0.305 | -0.262 | 0.274 | -0.555 |
| (0–2) | p-value | 0.023 | 0.108 | 0.102 | 0.111 | 0.006 |
| | n | 36 | 29 | 40 | 35 | 23 |

* Spearman rank (non-parametric) correlation.

DB composite score. There were no significant differences in age, gender, education or ante-mortem clinical diagnosis between participants with (n = 23) and those without (n = 18) DBs data available to create the composite measure.

The composite z-score is normally distributed; K-S goodness of fit test D(23) = 0,13; p>0.15. The reasons for missing DBs included the in-home sensor technology being removed for various reasons (e.g., participant moved from independent living to assisted living) or the participant being hospitalized for the last several months of his or her life. These patients continued to be clinically followed, but did not have sensor data for that time period, which resulted in a gap between the last sensor data and death. In order to determine the potential impact of this gap in data collection, Spearman rank correlations were rerun removing outliers, defined as participants with greater than 2 years between sensor data collection and death (remaining n = 17); all results remained significant. Participants' antemortem clinical diagnoses, based on clinician evaluation at last research visit prior to death, were: cognitively normal (46%), MCI (39%), and dementia (15%). Causes of death was available for 40 participants, and included cardiovascular-related (n = 15), pneumonia/inanition (n = 9), cancer (n = 8), unknown (n = 4), acute organ failure (n = 2), and suicide (n = 2).

On neuropathological evaluation, no participants were Braak stage zero. For the statistical analysis, Braak stages were categorized into three groups: I/II (n = 7), III/IV (n = 29), and V/VI (n = 5). Among this cohort, 83% were found to have Braak stage III or higher NFTs on autopsy. Twenty percent (n = 8) were found to have moderate/frequent neuritic plaques while 80% had none or sparse neuritic plaques (**Table 1**). Other pathologies were relatively infrequent: large vessel stroke or lacunar stroke (17%), hippocampal sclerosis (5%) and Lewy bodies (7%).

The DB composite score significantly predicted NP severity ($R^2$ = 0.36, $F_{(2, 20)}$ = 5.66, $p$ = 0.01). **Fig 2A**; **S1 Table**), but not Braak staging ($R^2$ = 0.14, $F_{(2, 20)}$ = 1.68, $p$ = 0.21; **Fig 2B**). In the model examining DB composite score by NP severity, while those with sparse plaques (Beta = -0.43, SE = 0.26, t = -1.68, $p$ = 0.11) were not significantly different than the control group (no plaques), those with moderate/frequent neuritic plaques had a significantly lower / poorer DB composite score (Beta = -0.90, SE = 0.27, t = -3.34, $p$<0.01). Global cognition at death (as measured by latest annual MMSE score) did not discriminate between NP severity ($R^2$ = 0.09, $F_{(2, 38)}$ = 1.90, $p$ = 0.16) or Braak stages ($p$ = 0.42). Functional status (as measure by last FAQ score) also did not discriminate between NP severity or Braak stages.

When the postmortem pathology variables were treated as ordinal variables, higher (worse) Braak stage was significantly correlated with fewer number of days with computer use ($\rho$ = -0.438, $p$ = 0.018) and more total time in bed ($\rho$ = 0.395, $p$ = 0.019; **Table 2**). Higher (worse) NP severity was significantly correlated with slower walking speed ($\rho$ = -0.379 $p$ = 0.023) and a

lower DB composite score ($\rho$ = -0.555 p = 0.006). Braak score and plaque severity were not different among those with computer use (n = 29) DBs and those without (n = 12). The DB composite score did not significantly differ between participant groups with or without post mortem evidence of infarction or stroke (n = 7 with infarction; t(6) = -0.57, $p$ = 0.57). Other pathologies noted above (hippocampal sclerosis and presence of Lewy Bodies) were too infrequent within the sample to be engaged in further analysis.

## Discussion

The current exploratory study provides the first data examining correlations between digital biomarkers (DBs) of daily functioning and neuropathological markers in an aging cohort, extending beyond established associations of DBs with clinical diagnoses [19] and providing a potentially important keystone in examining decline in older adults via passive monitoring.

A composite of DBs of daily function, as well as individual DBs, were correlated with neuropathological findings even in individuals whose cognition was not significantly impaired at the time of measurement. These correlations were not present between MMSE and Braak stage, which has been reported in other studies [46]. This lack of correlation in our study is likely a reflection of the predominantly low to intermediate stage of neurofibrillary tangle pathology in our sample. Specifically, the majority (70.7%) of the participants in the current study were in an intermediate, Braak stage III/IV, with only 5% in Braak stage V/VI. Other studies which have examined the relationship of Braak stage to MMSE have also not found a relationship between MMSE and Braak stage III/IV [47].

Taken together, these preliminary results suggest DBs may be more sensitive at detecting neuropathological findings than commonly used cognitive screeners, self-report questionnaires, or clinical diagnosis, with the potential to provide useful information in clinical and research settings. Thus, DBs, particularly DB composite metrics, may hold significant promise in detecting incipient behavioral or functional changes in AD. However, the present cross-sectional findings require additional longitudinal studies in order to confirm these findings and importantly, to determine the trajectory and timing of DB changes relative to underlying neuropathologic change. Given the growing availability of in vivo AD neuropathological biomarkers (blood-based and imaging), the correlation between DBs and early AD pathologic change during life is suggested as a promising future avenue of study to substantiate the clinical utility of these DBs to reflect early stage AD pathology.

Although we identified specific DBs which were significantly correlated with one neuropathology but not the other (e.g., computer use with Braak stage but not NP severity), in general the sample sizes of the individual groups available for analysis limit the ability to make definitive statements about these relationships at this time. Nevertheless, we note that the neuropathologic change in this sample was not severe nor extensive; 80% had none or sparse neuritic plaques, and 88% were below Braak stages V/VI (neocortical neurofibrillary tangle involvement). Thus, these DB observations have been made in older adults with relatively mild to moderate pathological change consistent with the current conceptualization of amyloid and tau accumulation likely occurring well before the presence of functional and cognitive changes are detected with conventional clinical tests [5,48,49]. Changes in specific DBs that may preferentially utilize a number of brain networks, are likely to reflect disruption of these networks as the complex, slowly evolving, and regionally progressing neuropathological process plays out over time. Thus for example, the interplay of tau or neurofibrillary change with amyloid deposition would be expected to lead to possible bidirectional effects on sleep behaviors where there is a balancing between amyloid and tau aggregation [50] that depending on the timing

and distribution of these processes, may result in disturbed sleep that is manifested by time in bed or other measures such as restlessness [51] or sleep efficiency [52].

NFT count is a stronger predictor of functioning than amyloid accumulation [53,54], which may be reflected in the present finding of decrease in the cognitively demanding task of computer use correlated with Braak staging. The relationship between specific DB and neuropathology type is worth consideration of exploration in future studies.

This study includes several limitations that can be addressed in future studies. First, the diversity of the sample is limited in terms of race, gender, and educational attainment. Second, while DBs have been shown to potentially yield clinically significant outcomes in longitudinal studies with relatively small samples [37], the sample size of the reported study is small and findings should thus be considered preliminary. A larger sample would have allowed for additional analyses, such as examining the effect of potential covariates and important predictors of cognitive decline that could be investigated further in the context of the noted DBs, including but not limited to family history of neurodegenerative disease, cardio- and cerebrovascular risk factors, APOE genotyping, and polypharmacy. It is also possible that some DBs have a more complex relationship with daily functioning than examined in the current models. Future DB-pathological correlational studies may consider these alternative models and consider changes in home-based activity measured as intradaily stability, variability, as well as spatio-temporal extent captured over time [55–57].

Third, the obtainment of DBs requires several factors that may limit accessibility, such requiring participants to have reliable internet, which may be problematic in some rural settings. Fourth, staging of AD pathology is dependent on the utilized neuropathological scales. This study used the Braak and CERAD systems, which is a combination recommended by the National Institute on Aging and Reagan Institute. However, staging may vary should investigators use the Poly-Pathology AD assessment (PPAD9), which focuses more intently on cytoarchitectural disorder and gliosis, microvacuolization, and degree of neuronal degeneration in nine cerebral areas, along with NPs and NFTs [58].

Overall, findings of this novel study suggest that DBs of daily function hold potential to serve as behavioral proxies for assessing pre-dementia pathological findings. In the context of suboptimal conventions for early detection of cognitive dysfunction, functional decline, and clinical diagnosis, DBs may bridge an important gap in the detection and treatment of neurodegenerative processes in pre-dementia phases.

## Supporting information

**S1 Table.** a. Linear regression model showing association between DB composite score and Braak stages (n = 23). b. Linear regression model showing association between DB composite score and Neuritic plaque severity (n = 23).
(DOCX)

## Author Contributions

**Conceptualization:** Nathan C. Hantke, Jeffrey Kaye, Chao-Yi Wu, Hiroko H. Dodge, Zachary Beattie, Randy Woltjer.

**Data curation:** Nora Mattek.

**Formal analysis:** Nora Mattek, Randy Woltjer.

**Funding acquisition:** Jeffrey Kaye, Hiroko H. Dodge.

**Investigation:** Jeffrey Kaye, Hiroko H. Dodge, Randy Woltjer.

**Methodology:** Jeffrey Kaye, Nora Mattek, Zachary Beattie.

**Project administration:** Jeffrey Kaye, Nora Mattek, Zachary Beattie.

**Writing – original draft:** Nathan C. Hantke, Jeffrey Kaye, Nora Mattek, Chao-Yi Wu, Hiroko H. Dodge, Zachary Beattie, Randy Woltjer.

**Writing – review & editing:** Nathan C. Hantke, Jeffrey Kaye, Nora Mattek, Chao-Yi Wu, Hiroko H. Dodge, Zachary Beattie, Randy Woltjer.

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
