## [Decision Letter · Decision Letter 0]

15 Feb 2023

PONE-D-23-00721Correlating continuously captured home-based digital biomarkers of daily function with postmortem neurodegenerative neuropathologyPLOS ONE

Dear Dr. Hantke,

Thank you for submitting your manuscript to PLOS ONE. After careful consideration, we feel that it has merit but does not fully meet PLOS ONE’s publication criteria as it currently stands. Therefore, we invite you to submit a revised version of the manuscript that addresses the points raised during the review process.

ACADEMIC EDITOR: After careful consideration by 3 Reviewers and an Academic Editor, all of the critiques of all three Reviewers must be addressed in detail in a revision to determine publication status. If you are prepared to undertake the work required, I would be pleased to reconsider my decision, but revision of the original submission without directly addressing the critiques of the 3 Reviewers does not guarantee acceptance for publication in PLOS ONE. If the authors do not feel that the queries can be addressed, please consider submitting to another publication medium. A revised submission will be sent out for re-review. The authors are urged to have the manuscript given a hard copyedit for syntax and grammar.

We look forward to receiving your revised manuscript.

Kind regards,

Stephen D. Ginsberg, Ph.D.

Section Editor

PLOS ONE

Journal Requirements:

"Competing Interests: JK in the past 36 months has been directly compensated for serving on a Data Safety Monitoring Committee for Eli Lilly, the Scientific Advisory Board of Sage Bionetworks, the Roche/Genentech Scientific Advisory Committee for Digital Health Solutions in Alzheimer’s Disease, and as an external Advisory Committee member for two Alzheimer's Disease Research Centers. He has received research support awarded to his institution (Oregon Health & Science University) from the NIH, NSF, the Department of Veterans Affairs, USC Alzheimer’s Therapeutic Research Institute, Merck, AbbVie, Eisai, Green Valley Pharmaceuticals, and Alector.  He holds stock in Life Analytics Inc. for which no payments have been made to him or his institution. He has received reimbursement through Medicare or commercial insurance plans for providing clinical assessment and care for patients. He has served on the editorial advisory board and as Associate Editor of the journal, Alzheimer's & Dementia and as Associate Editor for the Journal of Translational Engineering in Health and Medicine. HD works as a consultant for Biogen and is supported by the following federal grants: NIH R01AG051628, R01AG056102, R01AG069782, P30AG066518, R01AG072449, P30AG008017, P30AG024978, U2CAG054397, R01AG056712, R01AG0380651, P30AG053760, U01NS100611, U2CAG057441, U01NS106670, R01AG054484, RF1AG072449. ZB and JK have a financial interest in Life Analytics, Inc., a company that may have a commercial interest in the results of this research and technology. This potential conflict of interest has been reviewed and managed by Oregon Health & Science University. Remaining authors have no disclosures."

Reviewers' comments:

Reviewer's Responses to Questions

**Comments to the Author**

1. Is the manuscript technically sound, and do the data support the conclusions?

Reviewer #1: Partly

Reviewer #2: Yes

Reviewer #3: Partly

2. Has the statistical analysis been performed appropriately and rigorously? 

Reviewer #1: No

Reviewer #2: Yes

Reviewer #3: Yes

3. Have the authors made all data underlying the findings in their manuscript fully available?

Reviewer #1: No

Reviewer #2: Yes

Reviewer #3: Yes

4. Is the manuscript presented in an intelligible fashion and written in standard English?

Reviewer #1: Yes

Reviewer #2: Yes

Reviewer #3: Yes

5. Review Comments to the Author

Reviewer #1: In this article the authors correlate some digital biomarkers of activity of daily living with disease stage in patients with Alzhimer's disease with postmortem markers of pathology. While small in size and scope, I think the paper has merits.

Comments:

- there is basically only one meaningful results, that is correlation of composite score with neuritic plaques. It appears the analysis involved a global F-test of significance and the additional Wald's tests for the coefficients. If so it seems to me that figure 2 can be made to include all of the information and table 2 moved to supplementary. Even if it's not significant, I think a boxplot of composite stratified by Braak is also worth showing.

- As I mentioned, there is only one meaningful result. Which I find puzzling, since figures 3 and 4 suggest very clear patterns for individual subscores. I believe it possible that other significant results may be recovered with more powerful statistical methods. One thing that can easily be attempted is to exploit the fact that both postmortem markers are ordered variables, which means they can be treated as quantitative variables and save one degree of freedom. Also I believe that the assumption of normality may be faulty here. I guess the authors did test for normality, but normality tests like KS are greatly overrated, as they are conservative with small samples, which is exactly when they are most needed. It is usually more sensible to look at the actual distribution and choose analysis method based on the data generating process. For example, days spent at computer is obviously count data, those are usually Poisson distributed, and the fact that SD is basically equal to the mean reinforces this assumption as that's a characteristic of the Poisson distribution. A quasi-poisson generalized linear model may be a better call for that specific biomarker. Time spent in bed and time spent not at home are time variables, those usually display skewed distributions and are better modeled after a logarithmic transformation.

I suggest to refine the analysis to verify if those results really are non-significant.

- In any case, figures 3 and 4 do not work very well; the numbers above the columns are redundant, what is needed and is missing is some measure of variation. Please add error bars, or switch to a boxplot for consistency with the composite data.

Reviewer #2: This is a novel study describing the relationship between a passive digital biomarker composite and AD pathology on autopsy. The main issues with the paper are the small sample size and missing data within the collected sample.

Specifically, only a proportion of the participants had sufficient data to allow the generation of the composite. This is significant as the individual variables did not associate significantly with the AD outcomes. In addition, a significant proportion of the sample died while the study was taking place. This is somewhat surprising given that the participants at baseline had no significant medical comorbidities (inclusion) - it is unclear how this information was recorded and its reliability.

A key point made by the paper is that participants did not have dementia at baseline. However, this appears not to have been explored to the extent that is deemed standard in the dementia literature e.g. Clinical Dementia Rating scale. Instead, the authors relied on a MMSE cut-off score for diagnosing dementia which is considered poor practice in the dementia literature. The authors then proceed to report MCI incidence at death but it is unclear how many participants met this criterion at baseline. Perhaps not surprisingly, a significant proportion were diagnosed subsequently either with MCI (39%) or dementia (15%) - it is unclear how and when this information was obtained or the criteria for diagnosing either condition. Presumably, this was done clinically and thus not using a MMSE cut-off only.

It is also unclear how long participants had the digital technology in their homes (2008-2018 was mentioned as the study duration). This is an important component as it needs to be taken into account when justifying the decision to only look at the data from a period separated from death by 12 months. How long was the mean duration of recording in the study? What was the gap between the data analysis window and baseline? Also, looking at the range of time between last recording and death, there is a large gap (1 day to 90 months, the median being 1 day). This is a very large variation with likely a number of outliers who had years in between the recording and the pathology assessment. In addition, it is unclear why there was such a variability in terms of availability of digital biomarker data across the cohort.

APOE4 status was recorded but not used in the analysis. Please justify.

A minor point is that Table 1 is confusing e.g. hippocampal sclerosis (n=19) - does this mean that only 19 were assessed for presence of hippocampal sclerosis? FAQ abbreviation needs to be disambiguated

Reviewer #3: Thank you for allowing me to review this interesting paper. Nice methodology with a lot of real world potential and interesting results.

Abstract

‘Sensitive to early cognitive change ‘ -- this is too much of a claim as neither the cited references in the introduction nor the manuscript itself is showing this

Page 4: High data-capture frequency passive sensors provide digital biomarkers [DBs] of objective change in daily function (13) --- I don’t think reference 13 is backing up this claim.

Page 5: provides insight into every day cognitive function and can predict conversion to Mild Cognitive Impairment (MCI) prior to a clinical diagnosis (14-19). --- Several of the references refer to online surveys and not digital biomarkers. Reference 19 is referring to a study to validate DB, but is not yet providing empirical evidence.

Page 5 - A prior study of the decline in computer use over time found a

significant relationship between decreasing computer use and medial temporal lobe atrophy as

determined with volumetric MRI (20) --- Reference 20 is referring to a cross-sectional study, whereas the authors imply that results are longitudinal (decline … over time).

Introduction is severely lacking with respect to how and what references are used. Several sentences are not clearly backed up by evidence as references are actually not fitting to the claims.

Page 7 -- unclear how the home sensor system is measuring out of home time, given that there are two people living in the home. References are unclear how the system works in such scenarios. The authors should provide more information and clarify how they ensure that they measure time out of home for the participant only.

Page 8 -- ‘early markers of change’ -- the authors are not doing a longitudinal study, but purely cross-sectional, hence speaking about ‘change’ is misleading

Page 8 -- Figures 3 and 4. Barplots are not acceptable statistical representations of between group differences. Please change these figures to boxplots.

Page 8 -- ‘(last MMSE before death) and a functional measure (FAQ at last visit before death)” . following the arguments of the authors about average the full year of DB before death to avoid acute changes before death, here also only patients with MMSE and FAQ measures significantly before death (several month) should be included for a fair comparison

Page 9 -- ‘23 volunteers (56%) died while their home was sensored;’ should be removed as the next sentences offer better and cleare explanations.

Page 9 -- ‘cessation of computer use due to various difficulties’ - the authors should explain if these where potentially related to mental decline, as this would constitute a bias in their score/analysis (ceiling effect).

Methods: generally well explained. Two concerns

-- significant levels not explained. Seems the authors imply a 0.05 significance level cut-off. Is the study powered to detect anything there or not? Otherwise I would advise to be careful with the use of ‘significant difference’ and rather compare results in terms of p-value ranks etc. Otherwise I would expect e.g. in the discussion a comparison of the MMSE vs. NP results here (authors state it as ‘not significant’) vs. the ones found in other studies to put things into a general perspective.

-- methodology: why do the authors use linear regression, when they want correlations. Here spearman rank correlations could have been a better alternative to also avoid problems with normality assumptions.

Page 10: unclear what statistical methodology was used for the stroke analysis. Is it stroke vs. no-stroke? In that case I don’t see how a linear regression would work. Also I am concerned with normality assumption and in such cases use non-parametric testing if necessary.

Page 11: “The current study provides the first, preliminary data validating correlations between digital biomarkers (DBs)” -- I don’t agree with the term ‘validating’ . this is an exploratory first analysis. For ‘validation’ I would accept earlier findings to be referred to and a study that is powered to actually validate. Also, the authors are using linear regression, which is not ‘correlation’. The authors should use the correct statistical language here.

Page 11: “It is also possible that some DBs have a more complex relationship with functioning than presently captured in the current models.” -- I think instead of ‘with functioning’ the authors mean ‘with changes in the brain’. Their DBs are supposed to be be measures of ‘function’ itself, so there should not be any type of ‘relationship’.

The sentence ‘DBs, particularly DB composite metrics, may hold significant

promise in detecting incipient behavioral changes in AD, … ‘ → the authors just quote here other papers, but don’t contextualize this with their analysis. The authors should actually use this to contextualize the shortcomings of their study, in the sense that despite them having collecting longitudinal data over several years, they have not provided a longitudinal analysis. They have not shown reproducibility or replicability analysis either. And making claims on 90% sample size decrease are way too early.

Also, the authors are not discussing ‘cause and effect’ . Actually with their longitudinal, several years data, they could have provided more data for some suggestion in the direction. Is it lifestyle that could have an influence on brain deterioration, or vice versa? Here there is a tendency of a biased story teeling in there that it is ‘brain deterioration’ → ‘lifestyle changes’ → ‘MMSE changes’. I am not saying this is not possible, but in many diseases lifestyle changes are the last thing that changes as people (and their brains) have coping mechanisms.

6. PLOS authors have the option to publish the peer review history of their article (what does this mean?). If published, this will include your full peer review and any attached files.

Reviewer #1: No

Reviewer #2: **Yes: **Ivan Koychev

Reviewer #3: No

---

## [Author Response · Author response to Decision Letter 0]

9 May 2023

Nora

PONE-D-23-00721

Correlating continuously captured home-based digital biomarkers of daily function with postmortem neurodegenerative neuropathology

PLOS ONE

Reviewer #1: In this article the authors correlate some digital biomarkers of activity of daily living with disease stage in patients with Alzheimer’s disease with postmortem markers of pathology. While small in size and scope, I think the paper has merits.

Comments:

1. there is basically only one meaningful results, that is correlation of composite score with neuritic plaques. It appears the analysis involved a global F-test of significance and the additional Wald's tests for the coefficients. If so it seems to me that figure 2 can be made to include all of the information and table 2 moved to supplementary. Even if it's not significant, I think a boxplot of composite stratified by Braak is also worth showing.

We agree with the Reviewer’s comments regarding the Figures. We have created boxplots, labeled as Figure 2a & 2b, and moved Table 2 to supplementary. We also ran Spearman rank correlations, and have accordingly updated our findings. 

2. As I mentioned, there is only one meaningful result. Which I find puzzling, since figures 3 and 4 suggest very clear patterns for individual subscores. I believe it possible that other significant results may be recovered with more powerful statistical methods. One thing that can easily be attempted is to exploit the fact that both postmortem markers are ordered variables, which means they can be treated as quantitative variables and save one degree of freedom. Also I believe that the assumption of normality may be faulty here. I guess the authors did test for normality, but normality tests like KS are greatly overrated, as they are conservative with small samples, which is exactly when they are most needed. It is usually more sensible to look at the actual distribution and choose analysis method based on the data generating process. For example, days spent at computer is obviously count data, those are usually Poisson distributed, and the fact that SD is basically equal to the mean reinforces this assumption as that's a characteristic of the Poisson distribution. A quasi-poisson generalized linear model may be a better call for that specific biomarker. Time spent in bed and time spent not at home are time variables, those usually display skewed distributions and are better modeled after a logarithmic transformation. I suggest to refine the analysis to verify if those results really are non-significant.

Thank you very much for these analytic suggestions. Reviewer 3 recommended using correlations instead of using linear or generalized regression models. Since using different regression models according to the distributions of outcomes (e.g., Poisson models, linear regression, multinomial models) will give complexity in interpreting the results, we decided to provide correlations. We ran a Spearman rank (non-parametric) correlation with the biomarker activity metrics and postmortem pathology, data which is now presented in our new Table 2 in the manuscript. We have integrated the findings into the Results section (pg. 10):

“When the postmortem pathology variables were treated as ordinal variables, higher (worse) Braak stage was significantly correlated with fewer number of days with computer use (ρ = -0.438, p=0.018) and more total time in bed (ρ =0.395 , p=0.019; Table 2). Higher (worse) NP severity was significantly correlated with slower walking speed (ρ= -0.379 p=0.023) and a lower DB composite score (ρ= -0.555 p=0.006).”

In any case, figures 3 and 4 do not work very well; the numbers above the columns are redundant, what is needed and is missing is some measure of variation. Please add error bars, or switch to a boxplot for consistency with the composite data.

We agree with the reviewer that boxplots are more appropriate and have revised our Figures 3 & 4.

Reviewer #2: This is a novel study describing the relationship between a passive digital biomarker composite and AD pathology on autopsy. The main issues with the paper are the small sample size and missing data within the collected sample.

1. Specifically, only a proportion of the participants had sufficient data to allow the generation of the composite. This is significant as the individual variables did not associate significantly with the AD outcomes. In addition, a significant proportion of the sample died while the study was taking place. This is somewhat surprising given that the participants at baseline had no significant medical comorbidities (inclusion) - it is unclear how this information was recorded and its reliability.

The portion of participants who died during the study was expected and not atypical given the age of the cohort. The design of the study was for all subjects to be followed with in-home sensors until death, with the present data collected over a 10-year time period. The mean age of our participants at death was 92, which is older than the U.S. average of 76 years old reported by the CDC in 2022 (DOI: https://dx.doi.org/ 10.15620/cdc:118999). With regards to the reviewer’s first comment, we examined if there were fundamental differences in participants with (n=23) and without (n=18) the composite measure available. There were no differences found on age at death, gender, education or antemortem clinical diagnosis between groups. We have added this finding to the first paragraph of the results section (pg 9).

In this revision, we have provided more details on our methodology based on the reviewers’ comments. At entry, participants were determined to be in average health for their age, with well-controlled chronic diseases and comorbidities or none at all, assessed in the same manner as described in reference 12 including review of medical histories, medication lists, and completion of the modified Cumulative Illness Rating Scale (rating co-morbidities and health status). Medical illnesses with the potential to limit physical participation (e.g., wheelchair bound) or likely to lead to death over 35 months (such as certain cancers) were study exclusions. We have added this information to the Methods section of the manuscript (pg 6).

2. A key point made by the paper is that participants did not have dementia at baseline. However, this appears not to have been explored to the extent that is deemed standard in the dementia literature e.g. Clinical Dementia Rating scale. Instead, the authors relied on a MMSE cut-off score for diagnosing dementia which is considered poor practice in the dementia literature. The authors then proceed to report MCI incidence at death but it is unclear how many participants met this criterion at baseline. Perhaps not surprisingly, a significant proportion were diagnosed subsequently either with MCI (39%) or dementia (15%) - it is unclear how and when this information was obtained or the criteria for diagnosing either condition. Presumably, this was done clinically and thus not using a MMSE cut-off only. 

The reviewer’s comments have drawn our attention to the need for a description of the longitudinal visits, and we have expanded upon that section. All participants were cognitively within normal limits at baseline; no participants met criteria for MCI at that time. The MMSE was administered during the screening visit and the Clinical Dementia Rating scale (CDR) for the initial baseline visit and during all subsequent annual visits to define cognitive status and related diagnosis. Diagnosis of MCI and dementia was determined at each annual clinical evaluation, including CDR. Below is the revised description of our methods (pg. 6).

“Assessment of baseline health was based upon review of participants medical history, medication list, and completion of the modified Cumulative Illness Rating Scale (24, 25). Medical illnesses with the potential to limit physical participation (e.g., wheelchair bound) or likely to lead to death over 35 months (such as certain cancers) were study exclusions. All participants completed annual clinical evaluations and were followed until death, including administration of the Clinical Dementia Rating (CDR) scale (28) at initial and subsequent visits to monitor for the presence of MCI and conversion to dementia.”

3. It is also unclear how long participants had the digital technology in their homes (2008-2018 was mentioned as the study duration). This is an important component as it needs to be taken into account when justifying the decision to only look at the data from a period separated from death by 12 months. How long was the mean duration of recording in the study? What was the gap between the data analysis window and baseline? Also, looking at the range of time between last recording and death, there is a large gap (1 day to 90 months, the median being 1 day). This is a very large variation with likely a number of outliers who had years in between the recording and the pathology assessment. In addition, it is unclear why there was such a variability in terms of availability of digital biomarker data across the cohort. 

Study participants had sensors in their home for an average of 5.8 years (SD=2.4 years). The primary rationale for looking at DBs just from the 12 months prior to death was to capture trends in DBs, without focusing on terminal decline that would likely not be representative of the individual’s functioning in daily life. Twenty-three participants (56%) were monitored with in-home sensor technology up to death. Others had the in-home sensor technology removed for various reasons (e.g., moved from independent living to assisted living or nursing care) or were hospitalized for the last several months of his or her life. These patients continued to be clinically followed, but did not have sensor data for that time period which resulted in a gap between last sensor data and death. Regarding missing data for specific DBs, an example is that some participants quit using their computer for a variety of reasons and some participants experienced sensor failure (e.g., door sensors to monitor entering and leaving the home) that resulted in missing data. 

In order to determine the potential impact of the gap between last data collection and death, we reran the Spearman Rank correlation removing the 6 outliers defined as participants with greater than 2 years between sensor data collection and death (remaining n=17); all results noted in the manuscript remained significant. 

APOE4 status was recorded but not used in the analysis. Please justify. 

We agree that e4 status may be a variable of interest in future studies, but we feel it is outside the scope of our present analysis focused on neuropathology and DBs, particularly given our small sample. APOE e4 carrier status has been added as a descriptor in the first paragraph of the results section, but was not able to be included for a sub analysis. 

A minor point is that Table 1 is confusing e.g. hippocampal sclerosis (n=19) - does this mean that only 19 were assessed for presence of hippocampal sclerosis? FAQ abbreviation needs to be disambiguated. 

Yes, the hippocampal volume was assessed for 19 subjects of the large cohort. That is because we used the NACC neuropathology form and it was coded differently between versions. Hippocampal sclerosis was collected together with medial temporal lobe sclerosis in the Neuropathology Form version 9 and separately as its own variable in Neuropathology Form versions 10 & 11. Functional Activities Questionnaire is spelled out in the text, as well as in Table 1’s footnote.

We added to the manuscript (pg 8) that The NACC Neuropathology Form changed versions (versions 9, 10, and 11) over the course of the study, resulting in hippocampal sclerosis data only being available for a limited number of subjects (n=19).

Reviewer #3: Thank you for allowing me to review this interesting paper. Nice methodology with a lot of real world potential and interesting results.

Abstract

1. ‘Sensitive to early cognitive change‘ -- this is too much of a claim as neither the cited references in the introduction nor the manuscript itself is showing this. 

We tempered the language of the sentence by removing the comment on sensitivity to cognitive change.

2. Page 4: High data-capture frequency passive sensors provide digital biomarkers [DBs] of objective change in daily function (13) --- I don’t think reference 13 is backing up this claim.

Thank you. It is reference 12 that supports our statement of DBs measuring objective change; this has been changed in the manuscript. Lussier and colleagues (2018) conducted a systematic review (reference 12) which found 13 studies that use DBs to monitor objective change, including association of walking speed with MCI (Dodge et al., 2012) and computer usage with MCI (Seelye, et al., 2015). 

3. Page 5: provides insight into every day cognitive function and can predict conversion to Mild Cognitive Impairment (MCI) prior to a clinical diagnosis (14-19). --- Several of the references refer to online surveys and not digital biomarkers. Reference 19 is referring to a study to validate DB, but is not yet providing empirical evidence. 

To clarify this point, we have specifically defined in the introduction what is meant by a digital biomarker. With this in mind, the references noted in our manuscript focus on the relationships between patterns of computer usage and MCI, which reflects a passive measurement of function and supports our statement. Reference 14 refers to a study which found the pattern of older adult computer mouse movements was associated with MCI. Reference 15 is a study that looked at subjects’ time to complete online surveys at home with personal computing devices; in longitudinal analysis, individuals with MCI showed a pattern of changes in taking the survey not seen in those who were cognitively intact. This passive capture of metadata data around computer use (e.g., time to complete a survey, number of clicks) rather than conventional completion scores of a cognitive test online, represents a DB relevant assessing cognitive decline. Reference 16 expands upon this finding, showing patterns of computer usage are associated with future MCI diagnosis, showing subtle changes in DBs have the potential to be predictive of future clinical diagnosis. Reference 17 shows longitudinal changes in patterns of computer use and interaction is associated with MCI. We provided reference 18 to anchor our comment on “provides insight into every day cognitive function,” as it discusses the platform used within the manuscript, which we thought would be beneficial for our readers. We agree that reference 19 does not significantly support our statement and have removed it. (of note, our reference numbers have changed somewhat, but we used our original numbers in our response in order to be consistent with the reviewer’s comments).

4. Page 5 - A prior study of the decline in computer use over time found a

significant relationship between decreasing computer use and medial temporal lobe atrophy as

determined with volumetric MRI (20) --- Reference 20 is referring to a cross-sectional study, whereas the authors imply that results are longitudinal (decline … over time). 

We appreciate the reviewer bringing this to our attention. We agree, and have revised our sentence: “A prior cross-sectional study found a significant relationship between less daily computer use and medial temporal lobe atrophy as determined with volumetric MRI (19), a brain region that is known to be affected early-on in AD pathologically.”

5. Introduction is severely lacking with respect to how and what references are used. Several sentences are not clearly backed up by evidence as references are actually not fitting to the claims.

We have revised and strengthened our introduction to better reflect the purpose of the study and selected references that are more representative of our statements.

6. Page 7 -- unclear how the home sensor system is measuring out of home time, given that there are two people living in the home. References are unclear how the system works in such scenarios. The authors should provide more information and clarify how they ensure that they measure time out of home for the participant only. 

Only participants living alone were included in order to clearly disambiguate in-home movement. We have added this information to Digital Biomarker section of the methods (pg 7). 

7. Page 8 -- ‘early markers of change’ -- the authors are not doing a longitudinal study, but purely cross-sectional, hence speaking about ‘change’ is misleading. 

We have removed the statement of “early markers of change” and instead focused on cross-sectional interpretation.

8. Page 8 -- Figures 3 and 4. Barplots are not acceptable statistical representations of between group differences. Please change these figures to boxplots. 

Thank you, we appreciate the suggestion. Figures 2a, 2b, 3, and 4 are now all boxplots. 

9. Page 8 -- ‘(last MMSE before death) and a functional measure (FAQ at last visit before death)” . following the arguments of the authors about average the full year of DB before death to avoid acute changes before death, here also only patients with MMSE and FAQ measures significantly before death (several month) should be included for a fair comparison. 

Clinical visits with subjects, including administration of the MMSE and FAQ, only occurred once per year, with an average of time of 6.5 months between last clinical visit and death. While this resulted in some unavoidable variability among subjects, the DB data also focused on the last 12-month period of available data. In order to determine the potential impact of the gap between last data collection and death, we reran the Spearman Rank correlation removing the 6 outliers defined as participants with greater than 2 years between sensor data collection and death (remaining n=17); all results noted in the manuscript remained significant. We have added this statement to our results section (pg 10). 

10. Page 9 -- ‘23 volunteers (56%) died while their home was sensored;’ should be removed as the next sentences offer better and clearer explanations.

Thank you, we have removed the statement. 

11. Page 9 -- ‘cessation of computer use due to various difficulties’ - the authors should explain if these where potentially related to mental decline, as this would constitute a bias in their score/analysis (ceiling effect).

While the majority of volunteers were actively followed with in-home sensor technology up to death, some had the in-home sensor technology removed or discontinued for various reasons (e.g., moved from independent living to assisted living or nursing care) or were hospitalized for the last several months of his or her life, resulting in no DB data collection. We agree with the reviewer that it is entirely possible that cognitive decline is associated with decreased computer usage, which is supported by the finding of Braak staging being correlated with number of days of computer use (p=0.02) and prior publications showing computer declines as MCI progresses (Kaye et al., 2014). However, Braak score and plaque severity were not different among those with computer use (n=29) DBs and those without (n=12), which suggests that entire discontinuation of computer use is not biasing our findings, and all participants did not have difficulty using computers at the time of study enrollment. We added this information to our results section (pg 10).

12. Methods: generally well explained. Two concerns

-- significant levels not explained. Seems the authors imply a 0.05 significance level cut-off. Is the study powered to detect anything there or not? Otherwise I would advise to be careful with the use of ‘significant difference’ and rather compare results in terms of p-value ranks etc. Otherwise I would expect e.g. in the discussion a comparison of the MMSE vs. NP results here (authors state it as ‘not significant’) vs. the ones found in other studies to put things into a general perspective.

We have added additional information to our results section. The reviewer makes the observation that conventional wisdom and large cohort studies find a correlation of tau or tangles with cognition/MMSE, but we did not between MMSE and Braak stage. We believe this is because the post mortem data set in our study is skewed to low or only moderate levels of NFTs (few Braak Stage V/VI). Prior studies using a NACC database study (n=192) that looked at MMSE categories, obtained within 2 years of death, being associated with Braak Stage (with none/I/II being the reference against III/IV and V/VI). Although the study concluded that Braak Stage “predicts” lower MMSE, this was driven by the V/VI cases, which our study does not possess many of. There was no significant relationship of MMSE to Braak III/IV compared to the 0/I/II state in this study. We have edited this paragraph in the text to reflect this explanation. 

“These correlations were not present between MMSE and Braak stage which has been reported in other studies (46). This lack of correlation in our study is likely a reflection of the predominantly low to intermediate stage of neurofibrillary tangle pathology in our sample. Specifically, the majority (70.7%) of the participants in the current study were in an intermediate, Braak stage III/IV, with only 5% in Braak stage V/VI. Other studies which have examined the relationship of Braak stage to MMSE have also not found a relationship between MMSE and Braak stage III/IV (49).”

13. methodology: why do the authors use linear regression, when they want correlations. Here spearman rank correlations could have been a better alternative to also avoid problems with normality assumptions.

We would like to thank the reviewer for the suggestion. We ran a Spearman rank (non-parametric) correlation with the biomarker activity metrics and postmortem pathology, data which is now presented in Table 2, and moved the linear regression to a supplemental table. We have integrated the findings into the Results section (pg. 10).

14. Page 10: unclear what statistical methodology was used for the stroke analysis. Is it stroke vs. no-stroke? In that case I don’t see how a linear regression would work. Also I am concerned with normality assumption and in such cases use non-parametric testing if necessary.

Stroke vs no stroke was examined with an independent t-test. We have stated this more clearly in the methods and adjusted the findings on pg 10 to read as follows:

“The DB composite score did not significantly differ between participant groups with a history of stroke vs no stroke (n=7; t (6)= -0.57, p=0.57).”

15. Page 11: “The current study provides the first, preliminary data validating correlations between digital biomarkers (DBs)” -- I don’t agree with the term ‘validating’ . this is an exploratory first analysis. For ‘validation’ I would accept earlier findings to be referred to and a study that is powered to actually validate. Also, the authors are using linear regression, which is not ‘correlation’. The authors should use the correct statistical language here.

We agree with the reviewer’s assessment here and have changed the verb to “examining”. We have also changed our statistical analysis to a spearman correlation. 

16. Page 11: “It is also possible that some DBs have a more complex relationship with functioning than presently captured in the current models.” -- I think instead of ‘with functioning’ the authors mean ‘with changes in the brain’. Their DBs are supposed to be be measures of ‘function’ itself, so there should not be any type of ‘relationship’.

We apologize that we did not communicate well our intent with this statement. Accordingly, we have expanded discussion (pp. 11-12) around both the need in the future to consider alternative models of DB activity change measurement and analysis, as well as the how both the extent and severity of neuropathology may affect the results. 

17. The sentence ‘DBs, particularly DB composite metrics, may hold significant

promise in detecting incipient behavioral changes in AD, … ‘ → the authors just quote here other papers, but don’t contextualize this with their analysis. The authors should actually use this to contextualize the shortcomings of their study, in the sense that despite them having collecting longitudinal data over several years, they have not provided a longitudinal analysis. They have not shown reproducibility or replicability analysis either. And making claims on 90% sample size decrease are way too early.

The claim of reduced sample size is based on prior publications, but unnecessary for the purpose of our study (Dodge et al., 2015; Wu et al. 2021). We agree that more research in this area is needed, and it is our hope that our study looking at in-home activity and brain pathology leads to future longitudinal studies with larger prospectively assessed samples (please see also our comment below). We view the cross-sectional nature of our current data is an initial step and we have revised our discussion to express this thought. 

18. Also, the authors are not discussing ‘cause and effect’ . Actually with their longitudinal, several years data, they could have provided more data for some suggestion in the direction. Is it lifestyle that could have an influence on brain deterioration, or vice versa? Here there is a tendency of a biased story teeling in there that it is ‘brain deterioration’ → ‘lifestyle changes’ → ‘MMSE changes’. I am not saying this is not possible, but in many diseases lifestyle changes are the last thing that changes as people (and their brains) have coping mechanisms.

The reviewer here poses a very interesting and important question which we think is beyond the scope of the current study. In order to accrue the sample to conduct this clinical (DB) – pathological correlation study, we have followed the study participants for relatively long periods of time (up to 10 years prior to death), but of course used only the last 12 months of DB data prior to death for proper correlation of last life activity to end of life neuropathology. Having identified DB associations with AD neuropathologies in this study using the conventional paradigm or methodology to examine clinical-pathologic post mortem correlations, we look forward in follow-up studies, to being able to examine more closely the trajectory and timing of change of the DBs and other clinical measures that might lead to or best predict AD neuropathology during life (using in vivo pathologic markers), as well as with further end of life, autopsy data. We have added this comment to the discussion (p. 11 of the revised manuscript).

---

## [Decision Letter · Decision Letter 1]

24 May 2023

Correlating continuously captured home-based digital biomarkers of daily function with postmortem neurodegenerative neuropathology

PONE-D-23-00721R1

Dear Dr. Hantke,

We’re pleased to inform you that your manuscript has been judged scientifically suitable for publication and will be formally accepted for publication once it meets all outstanding technical requirements.

Kind regards,

Stephen D. Ginsberg, Ph.D.

Section Editor

PLOS ONE

**Comments to the Author**

1. If the authors have adequately addressed your comments raised in a previous round of review and you feel that this manuscript is now acceptable for publication, you may indicate that here to bypass the “Comments to the Author” section, enter your conflict of interest statement in the “Confidential to Editor” section, and submit your "Accept" recommendation.

Reviewer #1: All comments have been addressed

2. Is the manuscript technically sound, and do the data support the conclusions?

Reviewer #1: Yes

3. Has the statistical analysis been performed appropriately and rigorously? 

Reviewer #1: Yes

4. Have the authors made all data underlying the findings in their manuscript fully available?

Reviewer #1: No

5. Is the manuscript presented in an intelligible fashion and written in standard English?

Reviewer #1: Yes

6. Review Comments to the Author

Reviewer #1: (No Response)

7. PLOS authors have the option to publish the peer review history of their article (what does this mean?). If published, this will include your full peer review and any attached files.

Reviewer #1: **Yes: **Alberto Ferrari

---

## [Editor Report · Acceptance letter]

1 Jun 2023

PONE-D-23-00721R1 

Correlating continuously captured home-based digital biomarkers of daily function with postmortem neurodegenerative neuropathology 

Dear Dr. Hantke:

I'm pleased to inform you that your manuscript has been deemed suitable for publication in PLOS ONE. Congratulations! Your manuscript is now with our production department. 

Kind regards, 

on behalf of

Dr. Stephen D. Ginsberg 

Section Editor

PLOS ONE